# Evolution of China's interaction with Middle Eastern countries under the Belt and Road Initiative

**Junhua Chen**[1,2], **Xiaolu Yang**[1]*, **Meijun Wang**[1], **Min Su**[1]

1 School of Geographical Sciences, Southwest University, Chongqing, China, 2 Research Centre for Western Asia geography, Geographical Society of China, Chongqing, China

* yxl5011@email.swu.edu.cn

**Data Availability Statement:** All relevant data has been successfully uploaded to the figshare database, with the data DOI: 10.6084/m9.figshare.24270526. This ensures the widespread availability and transparency of the data.

## Abstract

The Middle East holds a critical strategic position in global politics, economy, and military affairs, serving as a pivotal hub for the advancement of the Belt and Road Initiative (BRI) through both land and sea routes. Since the proposal of BRI, China's cooperation with Middle Eastern countries has steadily deepened. Consequently, examining the evolution of China's interaction with Middle Eastern nations over the past decade is of paramount significance for future development. This study utilizes the GDELT database to construct formulas for measuring event impact and bilateral relationship intensity. It analyzes the temporal development and spatial patterns of China's interaction with Middle Eastern countries while also examining the types of interactive relationships between China and individual countries in the Middle East under the principle of reciprocity. The findings indicate that the overall interaction between China and Middle Eastern countries remains stable. Cooperative relationships have transitioned from a "single cooperation" approach to a "dual cooperation" model involving Iran and Saudi Arabia. Moreover, the development trajectory has shifted from an imbalanced "north-high, south-low" pattern towards equilibrium, resulting in a general decline in conflict relations and a decrease in inter-country disparities. The prevalent type of interaction between countries is characterized by balance.

## 1. Introduction

After the conclusion of the Cold War between the United States and the Soviet Union, China adopted a proactive stance and actively integrated into the Western-dominated international system. It strengthened its ties with the United Nations and actively engaged in various international treaty frameworks. Simultaneously, China initiated efforts to establish regional international frameworks in the Asia-Pacific, such as forming the Shanghai Cooperation Organization and launching the "ASEAN+3" mechanism involving China, Japan, and South Korea. In the 21st century, China joined the World Trade Organization, emerging as a significant member within the Western-led international system. Due to the impact of the global financial crisis in 2008 and the European refugee crisis in 2010, the dominant position of the United States and other major Western powers in the international system was weakened.

**Funding:** The authors received no specific funding for this work.

**Competing interests:** The authors have declared that no competing interests exist.

China gradually assumed the role of a "primary architect" in constructing the international system, with its global influence steadily increasing.

The Middle East is situated at the intersection of Asia, Europe, and Africa, hosting crucial maritime trade routes such as the Suez Canal, the Strait of Mandeb, and the Strait of Hormuz. It serves as a vital crossroads connecting the East and West, facilitating communication between the Atlantic and Indian Oceans, and holds significant strategic importance in global politics, economy, and military affairs [1]. Furthermore, the Middle East is a key intersection for the land-based Silk Road Economic Belt and the 21st Century Maritime Silk Road [2], playing a pivotal role in the advancement of the Belt and Road Initiative (BRI) in both land and maritime domains [3].

Since the 1990s, China's economic and trade ties with Middle Eastern countries have been increasingly close. The introduction of BRI in 2013 further deepened cooperation between China and ME, expanding beyond traditional sectors such as energy resources and infrastructure [4] to include emerging areas like the digital economy [5] and green energy [6]. By 2022, China had become the top trading partner for Middle Eastern countries [7], with half of its crude oil imports originating from the region [8]. However, as an extension of China's western border region, the Middle East experiences perennial instability, ongoing wars and conflicts, fluctuating nationalistic sentiments among Middle Eastern countries, challenges to elite governance by populism, and dramatic shifts in extremism and terrorism [9]. The cooperative and conflictual interactions between China and Middle Eastern countries not only affect bilateral economic and trade development and the smooth progression of BRI but also have implications for China's energy security and national interests.

The bilateral relationship and cooperation between China and the Middle East have been significant topics of research in the fields of geography, international relations, and economics, both domestically and internationally. Existing studies have primarily focused on the causes and impacts of conflicts in the Middle East [10–13]. Additionally, research has explored China's government attitudes toward the region, geopolitical aspects, and China's energy cooperation strategies [14–18]. Particularly, there has been a growing body of research on the Middle East region within the context of BRI [19]. Economic cooperation between China and the Middle East often intertwines with geopolitical research and extends to the field of national security [20–23]. The prospects of bilateral cooperation under the BRI have received widespread attention [3, 24, 25].

While most scholars have employed qualitative analysis to measure China's bilateral relations with the Middle East, exploring the subject from a macro and holistic perspective, there is a lack of research utilizing quantitative methods to analyze the interactive relationships between China and individual Middle Eastern countries. In evaluating geopolitical relationships, quantitative assessment methods have gradually improved, with studies employing event analysis and modified gravity models to measure and analyze China's geopolitical relationships with other countries and regions [26, 27]. However, few studies have attempted to separate the interactions between cooperation and conflict from bilateral relationships. With the Belt and Road Initiative proposed and implemented over the past decade, conducting a temporal and spatial study of the cooperative and conflictual interactions between China and Middle Eastern countries can provide valuable insights into the reasons and trends behind the changing relationships between the Middle Eastern countries and China as a key node in BRI. Furthermore, it can offer effective reference points for the future development.

With the continuous development of computer science and big data technology, news data has provided new support for quantitative research in fields such as geopolitical relationships and international politics [28]. Scholars have primarily pursued two main approaches in utilizing the GDELT database. The first approach involves constructing geopolitical relationship

networks based on GDELT event data to study the evolution of global or specific regional relationships [29–31]. The second approach entails quantitatively studying international relations based on the meanings and characteristics of GDELT data fields and aligning them with disciplinary theories, as well as conducting identification and analysis of event types within the research region [32, 33]. The evolving dynamics of China's interactions with Middle Eastern countries in terms of time and space can effectively reflect the progress and impact of BRI in the Middle East. The GDELT database provides comprehensive and clear records of political events, making it an excellent data source for studying interactions between nations.

Therefore, this paper explores the interactive relationship between China and Middle Eastern countries under BRI based on the GDELT database. It quantitatively characterizes the cooperative and conflictual interactions between China and Middle Eastern countries, along with their evolutionary trends, by constructing indices of event impact, bilateral relationship strength, and categorization of interactive relationship types under mutual responses. The aim is to assess the impact of BRI on bilateral relations and provide effective references for its construction and China's investment cooperation in the Middle East.

## 2. Data and methodology

### 2.1 Study area

The "Middle East" is a vague geographical concept without clear boundaries or a defined geographic range. Currently, the academic community distinguishes between a broad and a narrow definition of the Middle East. The broad definition includes the entire West Asia and North Africa region, ranging from Iran to Morocco. The narrow definition refers to the "Three Continents and Five Seas," specifically Turkey, Syria, Israel, Palestine, Lebanon, Cyprus, Iraq, Jordan, Yemen, Oman, Saudi Arabia, Qatar, the United Arab Emirates, Kuwait, Bahrain, Egypt, and Iran on the Iranian Plateau, totaling 17 countries and regions with a population of approximately 438 million (2020) and covering an area of about 3,700 square kilometers [34]. In this paper, we adopt the narrow geographical concept of the Middle East, excluding Palestine due to the lack of relevant data in the GDELT database. Therefore, the study area comprises 16 Middle Eastern countries excluding Palestine (Fig 1).

### 2.2 Data source

GDELT is an open-access global media database (*https://www.gdeltproject.org/*) that monitors various types of news from over 100 languages worldwide. The database is updated every 15 minutes and covers events from January 1, 1979, to the present. GDELT's event data is directional and can clearly represent the scale and impact of cooperation or conflict initiated by the acting country toward the receiving country, facilitating accurate analysis of interstate interactions. So far, the recorded event scales of cooperation and conflict align well with the interactions between countries and regions. The GDELT event database has 58 fields, including the Quad Class field, which indicates whether an event is cooperative or conflictual, the Goldstein Scale field, which represents the degree of cooperation or conflict for an event, and the Num Mentions field, which measures the number of times the event is mentioned in all the articles in the database. This serves as an auxiliary indicator for measuring the importance and influence of an event.

Event data occurring between 2013 and 2022, involving China as the actor and the 16 Middle Eastern countries as the recipients, as well as event data with the 16 Middle Eastern countries as the actors and China as the recipient, were downloaded, resulting in a total of 453,476 data points. The validity of the data was verified by matching the event types and Goldstein Scale with the Gordon Scale factor. After processing, 453,323 valid event data points were obtained, including 388,401 cooperative events and 649,22 conflict events.

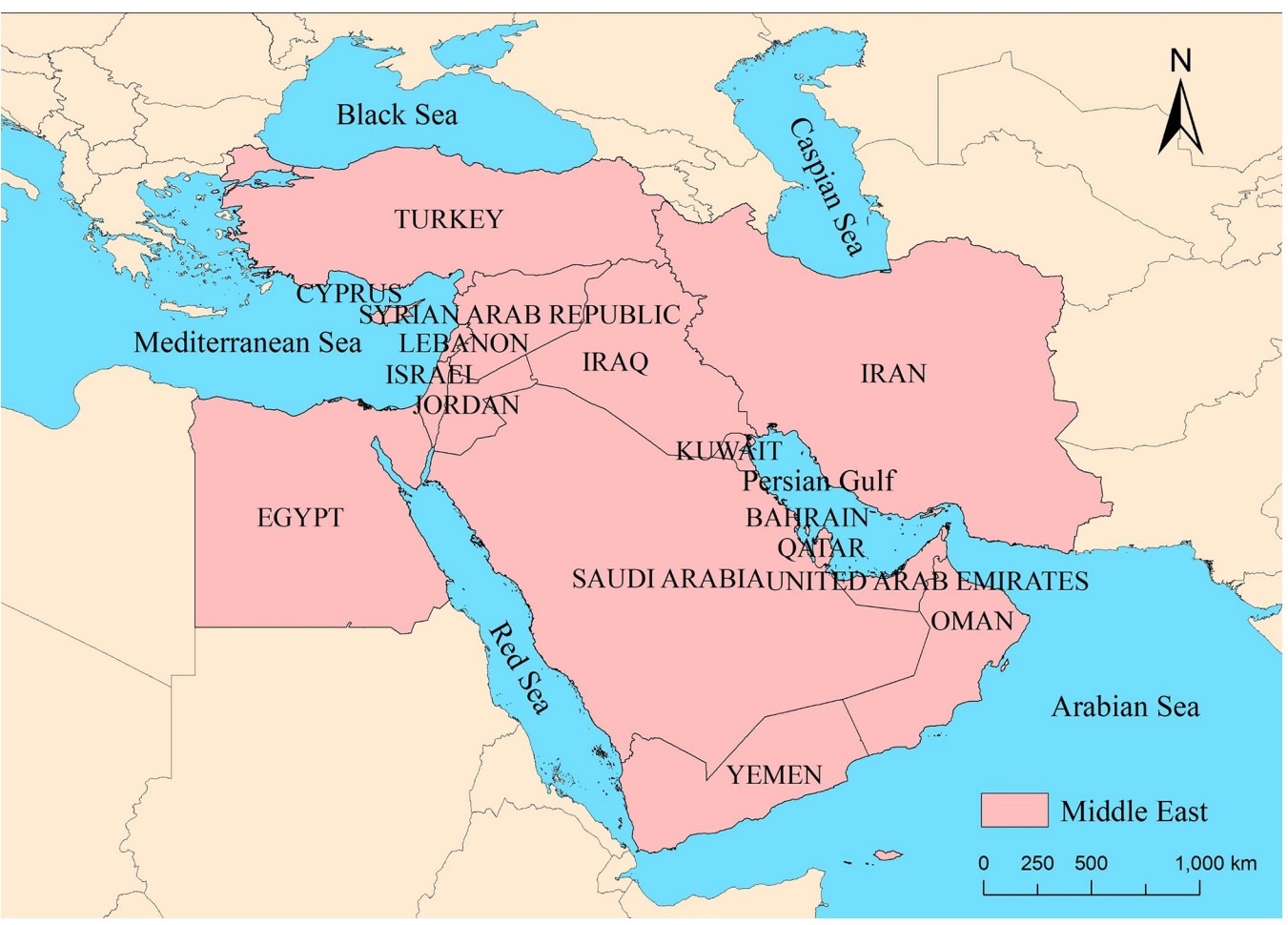

**Fig 1. The location of Middle East.** Note: Based on the standard map production with the map approval number GS(2016)1667 from the Chinese Ministry of Natural Resources(*http://bzdt.ch.mnr.gov.cn/*), the basemap remains unaltered.

## 2.3 Research methods

**2.3.1 Event impact.**  The Goldstein scores reflect the theoretical impact of events on the promotion or division of geopolitical relations between two countries and are commonly employed for quantitatively assessing political cooperation or conflict. Scholars often consider them as a significant indicator for studying the political and economic relationships between two nations [35, 36]. The event impact is based on the quantity of cooperation or conflict events, Goldstein scores, and the number of relevant news reports in the research area, representing the level of impact between countries in terms of cooperation or conflict events during the study period.

The formula for cooperation event impact is as follows:

$$I_{j,coo} = \sum_{i=1}^{n} G_{ij,coo} \times lg\left(\frac{\sum_{i=1}^{n} M_{ij,coo}}{n} + 10\right)$$

In the equation, $I_{j,coo}$ represents the impact of all cooperation events occurring in country $j$, representing China's cooperation relationship with country $j$ in the Middle East. $G_{ij,coo}$ is the Goldstein score of cooperation event $i$ occurring in country $j$, where a higher value indicates a

higher level of cooperation. $M_{ij,coo}$ represents the occurrence count of event $i$, which is the number of times the event is mentioned in all news articles in the database.

$lg\left(\frac{\sum_{i=1}^{n} M_{ij,coo}}{n} + 10\right)$ reflects the level of impact caused by event $i$.

The formula for conflict event impact is as follows:

$$I_{j,con} = -\sum_{i=1}^{n} G_{ij,con} \times lg\left(\frac{\sum_{i=1}^{n} M_{ij,con}}{n} + 10\right)$$

In the equation, $I_{j,con}$ represents the impact of all conflict events occurring in country $j$, representing China's conflict relationship with country $j$ in the Middle East. $G_{ij,con}$ is the absolute value of the Goldstein score of conflict event $i$ occurring in country $j$, where a larger absolute value indicates a higher level of conflict. $M_{ij,con}$ represents the occurrence count of event $i$, and

$lg\left(\frac{\sum_{i=1}^{n} M_{ij,con}}{n} + 10\right)$ reflects the level of impact caused by event $i$. To facilitate analysis, -1 is

multiplied to $G_{ij,con}$ to ensure that $I_{j,con}$ is a positive value.

**2.3.2 Bilateral relationship degree.** The bilateral relationship degree is calculated based on the cooperation event impact and conflict event impact between China and Middle Eastern countries, representing the level of interaction between China and Middle Eastern countries during the study period. The calculation formula is as follows:

$$I_j = I_{j,coo} - I_{j,con}$$

In the equation, $I_j$ refers to the bilateral relationship degree between China and country $j$ in the Middle East. $I_{j,coo}$ and $I_{j,con}$ represent the cooperation event impact and conflict event impact between China and country $j$.

**2.3.3 Interactive relationship types under reciprocity.** Reciprocity is an important concept in the study of international relations, referring to the interactive pattern where "one country takes a certain action, leading to another country taking a similar action" [37]. Based on the theory of reciprocity, the interactive relationship between China and Middle Eastern countries can be classified into three types: cooperation-dominant, conflict-dominant, and balanced, based on the impact of cooperation and conflict events between China and Middle Eastern countries. This classification allows for a clear identification of the types and evolution of the interactive relationship between China and Middle Eastern countries. The specific calculation formula is as follows:

$$R_j = \frac{I_{j,coo}}{I_{coo}} - \frac{I_{j,con}}{I_{con}}$$

In the equation, $R_j$ represents the interactive relationship type between China and country $j$. $I_{j,coo}$ and $I_{j,con}$ respectively represent the impact of cooperation events and conflict events between China and country $j$. $I_{coo}$ and $I_{con}$ represent the sum of cooperation event impacts and conflict event impacts, respectively, between China and all countries in the Middle East. When $R_j > 0.01$, it indicates that the proportion of the impact of cooperation events between China and country $j$ is significantly higher than the proportion of the impact of conflict events between China and country $j$, in comparison to the overall proportions of cooperation events and conflict events in the Middle East. This signifies a cooperation-dominant interactive relationship between China and country $j$. When $R_j < -0.01$, it indicates a conflict-dominant interactive relationship between China and country $j$, meaning that the proportion of the impact of

conflict events is higher than the proportion of the impact of cooperation events. When -0.01 $\leq R_j \leq 0.01$, it signifies a balanced interactive relationship between China and country $j$.

## 3. Evolution of China's interaction with Middle Eastern countries

### 3.1 Overall stable interaction

From 2013 to 2022, the interaction between China and Middle Eastern countries remained relatively stable, with cooperation outweighing conflicts (Fig 2A). The cooperative relationship between China and Middle Eastern countries experienced ups and downs, but overall, it improved after the introduction of the Belt and Road Initiative in 2013 (Fig 2B). In January 2016, the Chinese government issued the "Arab Policy Paper," officially proposing a "1+2+3" cooperation model with energy as the core, infrastructure construction and trade and investment facilitation as the two wings, and focusing on new energy, aerospace, and nuclear power [38, 39].

Xi Jinping's visits to Egypt, Iran, and Saudi Arabia emphasized China's new role in the Middle East, reaching the peak of cooperation between China and Middle Eastern countries in 2016. Subsequently, influenced by factors such as changes in the international situation, the cooperation between China and Middle Eastern countries gradually declined. In terms of specific countries, China's cooperation with Iran has consistently been the highest in the Middle East, with frequent exchanges and cooperation. Additionally, China has maintained a high

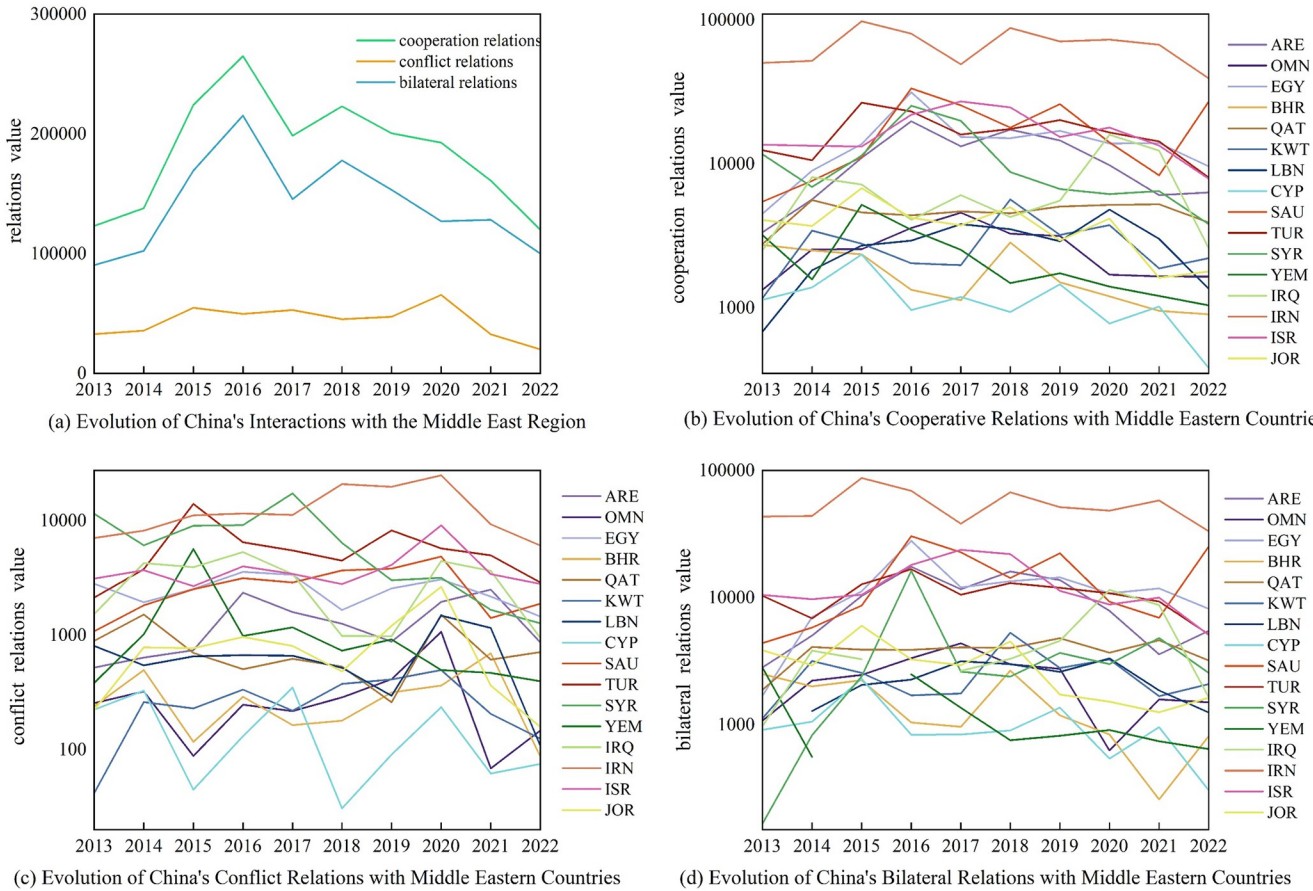

(a) Evolution of China's Interactions with the Middle East Region

(b) Evolution of China's Cooperative Relations with Middle Eastern Countries

(c) Evolution of China's Conflict Relations with Middle Eastern Countries

(d) Evolution of China's Bilateral Relations with Middle Eastern Countries

**Fig 2. Evolution of the temporal dynamics of China-Middle East country interaction relations.**

level of cooperation with major Middle Eastern countries such as Saudi Arabia, the United Arab Emirates, Egypt, Turkey, and Israel.

The overall conflict relationship between China and Middle Eastern countries showed a "fluctuating yet stable" pattern (Fig 2C). The conflict relationship reached its peak in 2020 due to factors such as the pandemic, economy, and politics, but it has decreased significantly in the past two years. In terms of specific countries, China's conflict relationship with Iran, Turkey, Syria, Israel, Iraq, and Egypt has been relatively high.

However, since the cooperative relationship between China and Middle Eastern countries outweighs the conflicts, the trend of bilateral relations is generally consistent with the cooperative relationship (Fig 2D). The Yemen data for 2015 (-457.25) and the Iraq data for 2016 (-1242.62) were negative, and logarithmic scales were used for plotting, therefore they are not displayed in the graph. In 2016, China issued official documents on cooperation with Arab countries, and Xi's visits to multiple countries in the Middle East resulted in the highest level of bilateral relations in recent years. Subsequently, bilateral cooperation steadily advanced, but due to unstable factors such as changes in the international situation and the COVID-19 pandemic, as well as limitations in the data collection of the GDELT event database itself, bilateral relations experienced some setbacks.

## 3.2 Transition from "single cooperation" to "dual cooperation"

China's cooperation in the Middle East has transitioned from a "single cooperation" model highlighted by cooperation with Iran to a "dual cooperation" model highlighted by cooperation with both Iran and Saudi Arabia (Fig 3). Iran has consistently been the country with the highest level of cooperation between China and the Middle East.

China and Iran have a long history of friendly exchanges spanning over two thousand years, from Zhang Qian's mission to the Western Regions during the Western Han Dynasty to the present. The initiation of the Belt and Road Initiative has presented new opportunities for collaboration between China and Iran. In an effort to alleviate pressure from the United States and the Western world, Iran is willing to engage in infrastructure development and other cooperative efforts within the framework of jointly building the Belt and Road Initiative with China. This collaboration aims to revitalize the domestic economy, fostering a closer and more resilient partnership between the two nations. In March 2021, China and Iran signed a comprehensive 25-year cooperation agreement encompassing political, strategic, and economic aspects, promoting the continuous improvement of the comprehensive strategic partnership between the two countries.

Saudi Arabia and Iran are both major powers in the Middle East and represent the Sunni and Shia branches of Islam, respectively, with significant influence in the region. After the "Arab Spring," Arab countries faced enormous pressure and challenges, and seeking better development became an urgent need for Saudi Arabia. The introduction of BRI has provided new possibilities for Saudi Arabia. Since President Xi Jinping's visit to Saudi Arabia in 2016, China and Saudi Arabia have established a comprehensive strategic partnership, continuously expanding cooperation in various fields and gradually forming a comprehensive and mutually beneficial cooperation pattern. By 2022, China has established a "dual cooperation" pattern with Iran and Saudi Arabia as the two major powers in the Middle East.

Saudi Arabia and Iran are both energy giants. Iran controls a crucial energy transport chokepoint, the Strait of Hormuz. Due to their vast energy reserves, strategic geographic locations, and religious significance, both countries hold important positions in the Middle East and even globally, making them crucial pivot countries for China's Belt and Road Initiative. However, due to historical reasons, Iran and Saudi Arabia have been involved in severe conflicts and contradictions, and they severed diplomatic relations for many years.

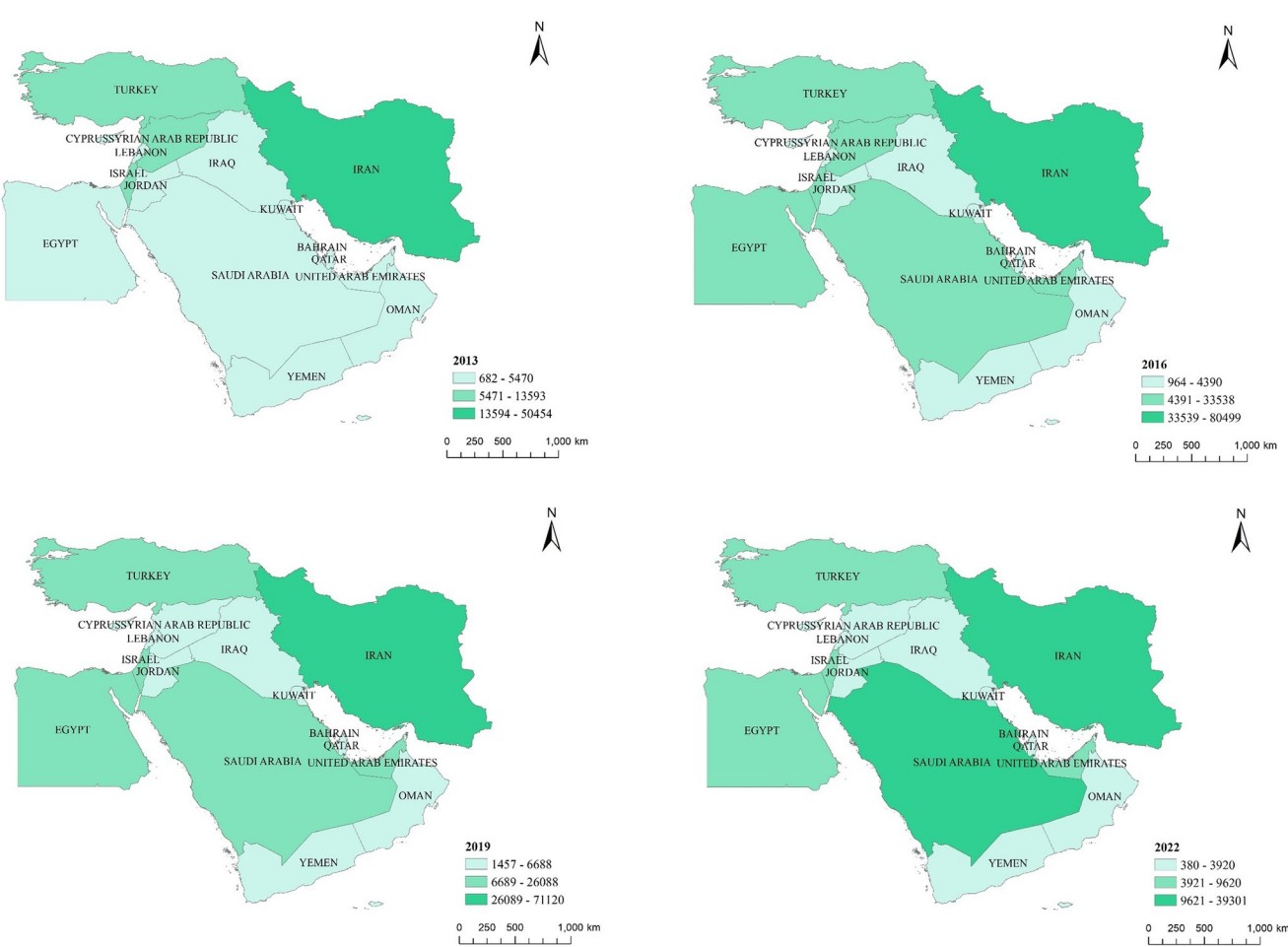

**Fig 3. Evolution of the spatial pattern of China-Middle East country cooperation relations.** Note: Based on the standard map production with the map approval number GS(2016)1667 from the Chinese Ministry of Natural Resources(http://bzdt.ch.mnr.gov.cn/), the basemap remains unaltered.

China, as one of the world's most important forces for justice, adheres to the principle of non-interference in other countries' internal affairs and maintains a neutral stance in its economic and trade cooperation and diplomatic exchanges with Iran and Saudi Arabia. In March 2023, with China's mediation, Iran and Saudi Arabia agreed to restore relations, and the three parties will work together to promote international peace and security. The reconciliation between these two major powers provides a new model for resolving various conflicts in the Middle East and offers the possibility of liberation from turmoil for countries like Yemen and Syria, while also providing strong impetus for the smooth advancement of BRI.

## 3.3 Development towards balanced cooperation in the "high north, low south" relationship

China's cooperation with countries in the Middle East has gradually shifted towards balanced development, moving away from the "high north, low south" relationship (Fig 3). In 2013, China's cooperation with northern countries such as Iran, Turkey, and Israel in the Middle East was significantly higher than with southern countries like Egypt and Saudi Arabia, resulting in noticeable regional disparities. However, in 2022, driven by BRI, China's cooperation with

various Middle Eastern countries has improved, and the spatial distribution has become more balanced.

China has maintained strong trade complementarity with Middle Eastern countries. Apart from the high export similarity between China and Turkey, the export similarity indices with the United Arab Emirates, Saudi Arabia, Iran, Qatar, Egypt, Bahrain, and Israel are relatively low. This favors the establishment of a trade-complementary division of labor [40]. In June 2014, Xi Jinping first proposed the concept of building a China-Arab community with a shared future and outlined the prospects for Belt and Road cooperation during the opening ceremony of the 6th Ministerial Conference of the China-Arab States Cooperation Forum, injecting political impetus into early cooperation projects between China and Middle Eastern countries. In early 2016, Xi visited three Middle Eastern countries (Saudi Arabia, Egypt, and Iran), where a total of 52 cooperation agreements were signed, covering various fields such as the Yanbu Petrochemical Complex project in Saudi Arabia, the Suez Economic and Trade Cooperation Zone project in Egypt, and the Tehran Metro project in Iran. China actively promotes economic diplomacy in the Middle East, participating in the construction of the Belt and Road Initiative and seeking collaborative growth [41]. Since 2016, China's cooperation relationship with Egypt and the Gulf Cooperation Council (GCC) countries, represented by Saudi Arabia and the United Arab Emirates, has gradually upgraded.

In 2019, China's cooperation relationship with Syria was downgraded. In December 2018, the United States announced its withdrawal from Syria, triggering intense power struggles among various factions, and extremist forces such as the Islamic State resumed their activities. The Syrian peace process faced new variables in 2019, and the situation in Syria, which had just shown signs of easing, once again plunged into turmoil. In 2020, the COVID-19 pandemic spread worldwide, hindering the development of economic, political, and trade cooperation between countries. Influenced by multiple factors, China's cooperation relationship with Syria declined. In January 2022, China and Syria signed a memorandum of understanding on Belt and Road cooperation, with the expectation that bilateral cooperation could take another step forward.

In 2022, China's cooperation relationship with Saudi Arabia was upgraded, forming a "dual cooperation" model in the Middle East. The United States has maintained a close informal alliance with Saudi Arabia for a long time, with oil being a crucial link in their relationship. However, the success of the U.S. shale oil revolution changed the international oil market landscape, turning both the United States and Saudi Arabia into sellers, leading to competition and strains in their relationship. In contrast, China's cooperation with Saudi Arabia has been upgrading in recent years, expanding from the traditional energy sector to infrastructure construction, new energy, high technology, aerospace, and other fields. BRI is deeply aligned with Saudi Arabia's "Vision 2030," making Saudi Arabia one of China's important partners in the Middle East.

## 3.4 Decrease in conflict relations and reduction of inter-state differences

From Fig 4, it can be observed that the countries with high values in conflict relations exhibit a high degree of similarity with those in cooperation relations, although the cooperation relations significantly surpass the conflict relations. As interaction and communication between the two sides increase, accompanied by a rise in the number of news reports, cooperation relations improve, with only a small minority opposing the cooperation that clearly promotes economic and social development in both countries.

The cooperation relationship between China and Iran has consistently remained at a high level, while the value of conflict relations also stands out in a horizontal comparison within the

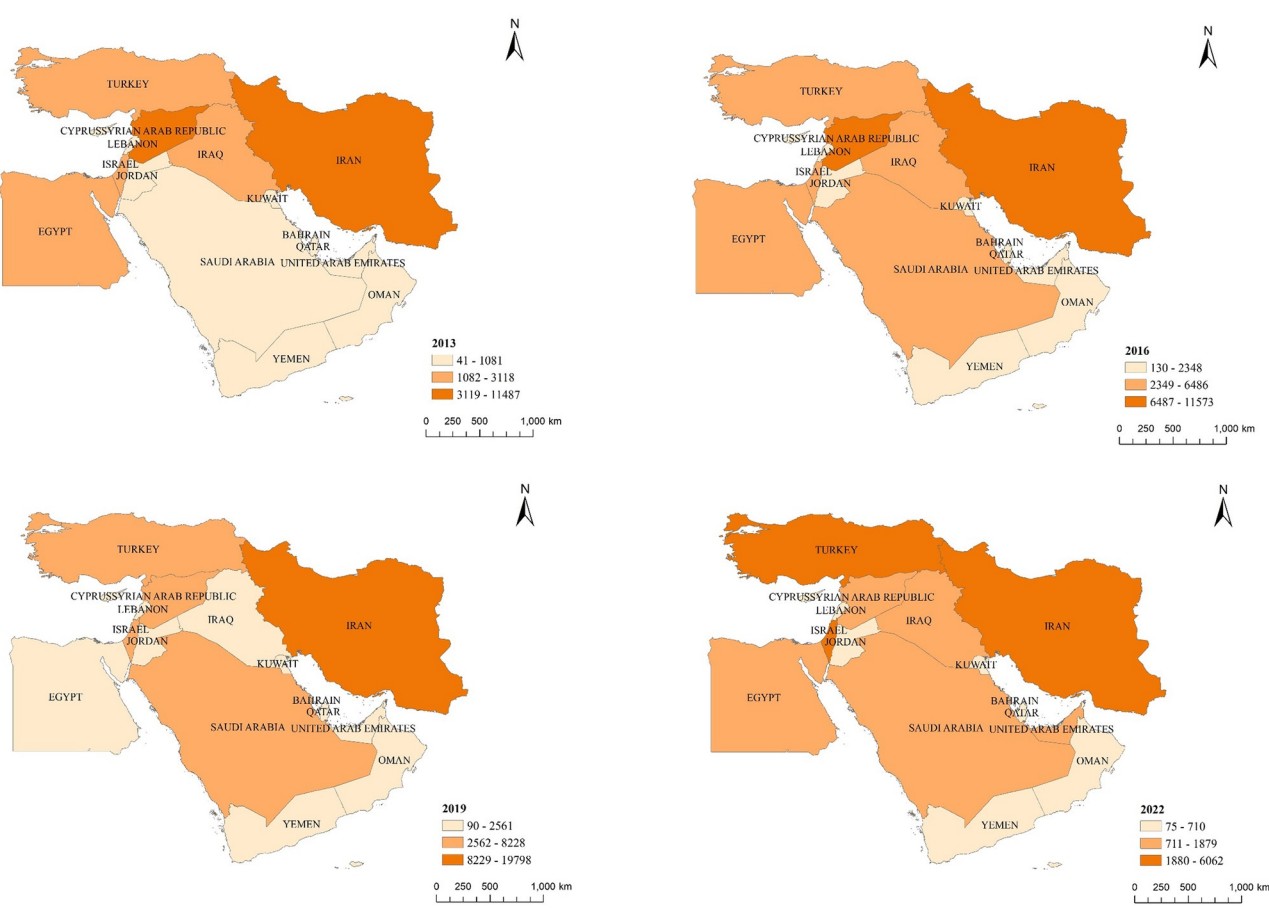

**Fig 4. Evolution of the spatial pattern of China-Middle East country conflict relations.** Note: Based on the standard map production with the map approval number GS(2016)1667 from the Chinese Ministry of Natural Resources(http://bzdt.ch.mnr.gov.cn/), the basemap remains unaltered.

Middle East region. Since 2016, China's cooperation relationship with Saudi Arabia has experienced a noticeable increase, accompanied by a similar change in conflict relations. Overall, from 2013 to 2022, the level of conflict between China and Middle Eastern countries has generally decreased, and inter-state differences have diminished.

In 2019, the conflict relations between China and Egypt, Iraq, and Syria were downgraded. After President Xi Jinping's visit to Egypt in 2016, the various cooperation agreements signed between China and Egypt gradually materialized, yielding fruitful results. People-to-people exchanges and cultural interactions have increased year after year. The benefits brought to Egypt by BRI are highly recognized by the government and the people. As a result, China-Egypt relations have continued to develop positively, leading to a significant decrease in conflict relations in 2019. However, with the outbreak of the COVID-19 pandemic, many economic, trade, and cultural exchange projects between China and Egypt could not proceed as planned, causing a significant impact on Egypt, whose pillar industries include oil, remittances, the Suez Canal, and tourism. In 2022, the conflict relations between China and Egypt showed a slight increase.

China established a strategic partnership with Iraq in 2015, but due to the volatile domestic situation in Iraq, many cooperations did not commence immediately. Starting from 2018, as the political situation in Iraq gradually stabilized and post-disaster reconstruction projects were initiated, China-Iraq cooperative projects have been implemented one by one, showing

positive momentum. The government and the people maintain a positive and friendly attitude towards the Belt and Road construction. Consequently, conflict relations decreased. However, in 2021, extremist forces resurfaced, posing an increased threat to Iraq's social security and stability. Coupled with the impact of the COVID-19 pandemic, many cooperation projects were interrupted, leading to an increase in conflict relations between China and Iraq in 2022.

Since the end of 2018, Syria has been affected by the intervention of multiple forces in its domestic situation after the United States announced its withdrawal. China adheres to an objective and just stance, refraining from involvement in Middle East conflicts or seeking interests in the region. As a result, China's cooperation relationship with Syria has been downgraded, accompanied by a decrease in conflict relations.

In 2022, the conflict relations between China and the United Arab Emirates (UAE) escalated, and the level of conflict with Israel intensified. The relationship between China and the UAE has always been an exemplar of global cooperation. Since establishing a comprehensive strategic partnership in 2018, the cooperation between both sides has continued to deepen. However, in 2021, the China-UAE relationship was somewhat affected by the COVID-19 pandemic, leading to an increase in conflict relations. Israel, known for its emphasis on technological development in areas such as computing, biotechnology, new materials, communications, and artificial intelligence, has engaged in numerous collaborations with China.

In 2020, with the endorsement of the United States, Israel persisted in advancing its territorial annexation plan. In May 2021, a large-scale conflict broke out between Palestine and Israel, leading to a stagnation in the peace process. Simultaneously, during the COVID-19 pandemic, there was a rise in unfriendly voices towards China internationally, contributing to the escalation of conflict relations between China and Israel.

# 4. China's interactions with Middle Eastern countries: Categorization

## 4.1 Cooperation-dominant type

During the period of 2013–2022, more than half of the Middle Eastern countries had a cooperation-dominant type of interaction with China. This means that the proportion of cooperative events with China outweighed the proportion of conflict events in terms of their impact. Compared to the overall situation in the Middle East, the interactions between these countries and China exhibited a cooperation-dominant pattern. According to Fig 5, the United Arab Emirates (UAE), Egypt, and Iran had cooperation-dominant interactions with China.

|  | 2013 | 2014 | 2015 | 2016 | 2017 | 2018 | 2019 | 2020 | 2021 | 2022 |
|---|---|---|---|---|---|---|---|---|---|---|
| ARE | 0.0103 | 0.0222 | 0.0353 | 0.0255 | 0.0355 | 0.0484 | 0.0534 | 0.0203 | -0.0436 | 0.0048 |
| EGY | -0.0549 | 0.0074 | 0.0133 | 0.0442 | 0.0116 | 0.0294 | 0.0280 | 0.0239 | 0.0161 | -0.0009 |
| IRN | 0.1802 | 0.1339 | 0.2275 | 0.0611 | 0.0280 | -0.0874 | -0.0827 | -0.0127 | 0.1160 | -0.0103 |
| SAU | 0.0093 | 0.0010 | 0.0021 | 0.0606 | 0.0723 | -0.0047 | 0.0459 | -0.0030 | 0.0059 | 0.1210 |
| OMN | 0.0026 | 0.0089 | 0.0097 | 0.0084 | 0.0189 | 0.0081 | 0.0067 | -0.0080 | 0.0080 | 0.0055 |
| BHR | 0.0145 | 0.0034 | 0.0083 | -0.0010 | 0.0025 | 0.0086 | 0.0005 | 0.0005 | -0.0167 | 0.0027 |
| QAT | -0.0064 | -0.0044 | 0.0073 | 0.0060 | 0.0114 | 0.0081 | 0.0195 | 0.0032 | 0.0126 | -0.0069 |
| KWT | 0.0081 | 0.0172 | 0.0081 | 0.0007 | 0.0057 | 0.0167 | 0.0069 | 0.0117 | 0.0050 | 0.0115 |
| LBN | -0.0206 | -0.0029 | -0.0003 | -0.0029 | 0.0063 | 0.0038 | 0.0079 | 0.0016 | -0.0189 | 0.0053 |
| CYP | 0.0020 | 0.0003 | 0.0096 | 0.0009 | -0.0009 | 0.0035 | 0.0053 | 0.0003 | 0.0043 | -0.0010 |
| JOR | 0.0259 | 0.0036 | 0.0157 | -0.0042 | 0.0031 | 0.0113 | -0.0120 | -0.0200 | -0.0018 | 0.0061 |
| ISR | 0.0092 | -0.0127 | 0.0083 | -0.0005 | 0.0694 | 0.0464 | -0.0141 | -0.0502 | -0.0280 | -0.0902 |
| IRQ | -0.0293 | -0.0679 | -0.0421 | -0.0963 | -0.0363 | -0.0036 | 0.0060 | 0.0129 | -0.0419 | -0.0311 |
| YEM | 0.0136 | -0.0188 | -0.0840 | -0.0075 | -0.0102 | -0.0103 | -0.0116 | -0.0005 | -0.0077 | -0.0133 |
| TUR | 0.0322 | -0.0349 | -0.1461 | -0.0486 | -0.0273 | -0.0254 | -0.0811 | -0.0044 | -0.0737 | -0.0939 |
| SYR | -0.2784 | -0.1307 | -0.1200 | -0.0961 | -0.2397 | -0.1086 | -0.0333 | -0.0179 | -0.0141 | -0.0388 |

**Fig 5. Classification of China's interactions with Middle Eastern countries.**

The UAE is located on the southern coast of the Persian Gulf in the Arabian Peninsula, with convenient transportation by sea, land, and air. It has well-developed infrastructure and a favorable business environment, making it one of the preferred destinations for China's "going global" strategy. Similarly, the UAE was one of the earliest countries to join the Belt and Road Initiative and has been China's largest export market in the Middle East. Over the years, the cooperation between the two countries has yielded fruitful results. The construction of the world's largest seawater desalination project by Chinese companies in Abu Dhabi's Taweelah has been successfully completed. In July 2018, Xi Jinping's visit to the UAE resulted in the establishment of a comprehensive strategic partnership and the enhancement of cooperation in various fields, propelling China-UAE relations to a new level. In 2021, China faced criticism and attacks internationally regarding its policies in combating the COVID-19 pandemic, which had a certain negative impact on China-UAE relations, leading to a conflict-dominant type of interaction. However, as comprehensive strategic partners, the deepening cooperation in pandemic response strengthened mutual friendship and political mutual trust between China and the UAE, restoring the bilateral relationship to a balanced type in 2022.

Egypt is located at a strategic land transportation hub between Asia, Europe, and Africa. The Suez Canal, an inland waterway in Egypt, connects the Atlantic Ocean and the Indian Ocean, giving it a superior geographical position. Egypt was one of the earliest Arab and African countries to establish diplomatic relations with the People's Republic of China. The foundation of cooperation between Egypt and China is strong, with close economic and cultural exchanges. In 2013, Egypt experienced political turmoil with ongoing conflicts between supporters and opponents of the Muslim Brotherhood, as well as significant tensions between the government and the public. The security situation was severe, leading to a conflict-dominant type of interaction between China and Egypt. In 2014, after President Abdel Fattah el-Sisi took office and the political situation gradually stabilized, he visited China and the two countries established a comprehensive strategic partnership, promoting long-term cooperation in politics, economy, trade, and culture. Since then, China and Egypt have maintained a friendly bilateral relationship, primarily characterized by cooperation-dominant interactions.

Iran guards the strategic Strait of Hormuz, an important oil transportation route, and possesses abundant oil and gas resources along with significant religious influence. It is also the country in the Middle East with the most comprehensive industrial system. Iran has long been an intimate friend of China in the Middle East, with inherent cooperation advantages in energy, infrastructure, high-tech industries, and other fields. The two countries have maintained a long-term friendly and pragmatic cooperative relationship. In 2016, Xi Jinping visited Iran, and the two countries officially established a comprehensive strategic partnership, embarking on a new journey of common development.

The Iran nuclear issue has always been a focal point of international concern. In 2015, Iran reached a comprehensive agreement on the Iran nuclear issue with the United States, the United Kingdom, France, Russia, China, and Germany. According to the agreement, Iran pledged to limit its nuclear program, and the international community lifted sanctions against Iran. Iran's economy showed positive development. In 2018, the United States unilaterally withdrew from the Iran nuclear agreement, escalating tensions between the US and Iran. The US imposed unilateral economic and financial sanctions on Iran, continuously expanding their scope. Under the new circumstances, China-Iran cooperation was affected. In 2021, with the rise of hardline factions, represented by President Ebrahim Raisi, Iran signed a comprehensive 25-year cooperation agreement with China, shifting the interaction between the two countries back to a cooperation-dominant type. In 2022, the Iran nuclear negotiations with the United States faced difficulties, and the international situation became turbulent and unstable, leading to a conflict-dominant type of interaction between China and Iran.

## 4.2 Balanced type

In the period from 2013 to 2022, there were Middle Eastern countries that maintained a balanced type of interaction with China, where neither cooperation nor conflict predominated. The interaction between these countries and China remained balanced compared to the overall situation in the Middle East. According to Fig 4, Saudi Arabia, Oman, Bahrain, Qatar, Kuwait, Lebanon, Cyprus, and Jordan had a balanced type of interaction with China.

Saudi Arabia, located on the Arabian Peninsula, borders the Persian Gulf to the east and the Red Sea to the west. It is one of the world's largest oil-exporting countries and has been a major supplier of crude oil to China for many years. Saudi Arabia has been the top recipient of Chinese investments among Arab countries in the past decade and is China's largest trading partner in West Asia and Africa. The two countries have numerous cooperative projects in infrastructure construction, investment, labor, agriculture, and other fields. Examples include the construction of the King Hamad Causeway and the light rail project in the holy city of Mecca. The friendly relations between the two countries have steadily developed. In 2016, Xi visited Saudi Arabia, and together with the Saudi King, they announced the establishment of a comprehensive strategic partnership. They signed multiple agreements and memoranda of understanding covering education, investment, technology transfer, industrial trade, and other areas. In 2022, Xi visited Saudi Arabia again, strengthening the alignment of BRI with the regional development strategy, and working together to build a community with a shared future for China and the Arab League in the face of significant global changes.

Oman, Bahrain, Qatar, and Kuwait, along with the UAE and Saudi Arabia, are members of the Gulf Cooperation Council (GCC) and are important oil-exporting countries in the world. They play a significant role in maintaining peace and promoting political and economic development in the Middle East. Since the establishment of the GCC, China has established connections with these countries and has been promoting the development of China-GCC bilateral relations with a friendly and cooperative attitude. Currently, China has become the largest importer and exporter among the GCC countries. Presently, China has become the largest importer and exporter within the Gulf Cooperation Council (GCC), fostering deepening Sino-Gulf relations with fruitful outcomes. Examples include cooperative projects like Oman's Ibri Photovoltaic Power Station, Kuwait's Mutlaa Large Rainwater Collection Basin, and ongoing negotiations for a China-Gulf Free Trade Zone. China's geopolitical-economic strategies in the GCC countries significantly enhance its economic influence in the Gulf region [42]. China and the GCC countries are important political, economic, and energy cooperation partners, and serve a model for China's cooperation with developing countries.

Lebanon, Jordan, and Cyprus are located on the eastern side of the Mediterranean Sea and are important nodes in the construction of BRI. They hold strategic positions. China established strategic partnerships with Jordan and Cyprus in September 2015 and November 2021, respectively, deepening cooperation under BRI and promoting economic development and national prosperity. Lebanon is experiencing political instability and a challenging security situation, with certain geopolitical risks. China has consistently provided assistance to Lebanon in maintaining national peace and security and, when conditions permit, encourages Chinese enterprises to participate in Lebanon's socio-economic development and promote the Belt and Road Initiative.

## 4.3 Conflict dominant type

Among the Middle Eastern countries, there were more countries with a conflict dominant type of interaction with China than those with a cooperative dominant type during the period from 2013 to 2022. This means that the proportion of conflict events between these countries

and China was significantly higher compared to the proportion of cooperative events, indicating a conflict dominant type of interaction between them. According to Fig 4, Israel, Iraq, Yemen, Turkey, and Syria had a conflict dominant type of interaction with China.

Israel pursues a US-centered imbalanced foreign policy, maintaining close ties with European countries while actively developing relations with emerging market countries such as China. In May 2013, China and Israel signed cooperation agreements in various fields including trade, investment, technology, education, and agriculture in Beijing. In March 2017, China and Israel jointly announced the establishment of a comprehensive innovation partnership, aiming to strengthen technological innovation and deepen cooperation in areas such as basic science, modern agriculture, clean energy, and biomedicine under the framework of BRI. In December 2019, the United States announced that it no longer considers Israeli settlements in the West Bank as "consistent with international law," leading to increased conflict between Palestine and Israel and heightened tension in the Middle East. In 2020, as the COVID-19 pandemic unfolded, there was an increasing international criticism towards China's epidemic prevention policies, leading to a shift in the bilateral relationship between China and Israel towards a conflict-driven approach.

The situation of war has persisted. Iraq has a long history of sectarian conflicts, with significant tensions between Shiite Muslims, Sunni Muslims, and Kurds. Since the Iraq War in 2003, the domestic political situation has been unstable, and the security situation has deteriorated continuously, leading to increased social divisions. The interference of terrorist forces has further complicated the situation, resulting in a conflict dominant type of interaction with China. At the end of 2017, Iraq achieved significant victories in the fight against terrorism, and in 2018, post-war reconstruction gradually began, with active participation in cooperation projects under BRI, such as the modernization of oil fields, leading to an improvement in bilateral relations with China. However, in 2021, extremist forces such as ISIS reignited the conflict, carrying out multiple terrorist attacks and violent clashes, hindering friendly relations between China and Iraq.

Yemen is located in the southwestern tip of the Arabian Peninsula, bordering the Red Sea, the Gulf of Aden, and the Arabian Sea. It is one of the least developed countries in the world. In 2014, the civil war broke out in Yemen, and the Houthi rebels seized the capital, Sanaa. Since then, Yemen has been plagued by ongoing conflicts between the government and the Houthi rebels, becoming a battleground for various factions and external powers.

Since the Justice and Development Party came to power in 2002, Turkey has gradually shifted its previously "pro-Western" foreign policy towards an eastward inclination, adopting a comprehensive geopolitical and proactive diplomacy. Turkey has sought to improve its relations with Middle Eastern countries, actively engaged in regional issues, and pursued political interests to enhance its geopolitical influence [43]. However, due to its high-profile involvement in the politics of other countries, Turkey's national security is to some extent influenced by the armed activities of the Kurdistan Workers' Party and extremist terrorist forces, leading to slow progress in cooperation with BRI. Turkey also has a collective memory associated with pan-Turkism, and many "East Turkistan" separatist elements have long operated in Turkey, distorting the truth and causing a cognitive bias among the Turkish population regarding China's policy in Xinjiang, resulting in anti-China sentiment and conflict-dominated China-Turkey relations.

In March 2011, the anti-government demonstrations sparked by the Arab Spring escalated into an armed conflict in Syria. Since then, Syria has been engulfed in ongoing warfare and has become a stage for international political maneuvering with the involvement of multiple external powers. The conflict between the Syrian government forces, opposition factions, and various extremist armed groups continues, while Turkey, Israel, Iran, the United States, Russia,

and other external powers engage in power struggles, leading to a continuous escalation of tensions. The peace process in Syria has been difficult and tumultuous. China has consistently adhered to the diplomatic principle of "the people of the Middle East are the masters of the Middle East" and has maintained a non-interference policy in the internal affairs of other countries. China supports a peaceful resolution to the Syrian crisis and has provided humanitarian assistance to Syria on multiple occasions.

## 5. Conclusion and discussion

This paper utilized the GDELT database to quantitatively analyze the temporal and spatial evolution of China's cooperation and conflict interactions with Middle Eastern countries from 2013 to 2022, accurately identifying the types of interactive relationships between China and various countries in the Middle East. The main conclusions are as follows:

China's interactions with Middle Eastern countries have remained stable, with a significant increase since the introduction of the Belt and Road Initiative (BRI).China has closer interactions with countries such as Iran, Egypt, Saudi Arabia, the United Arab Emirates, Turkey, and Israel.

The cooperative relationship has shifted from single cooperation with Iran to dual cooperation with both Iran and Saudi Arabia. The regional cooperative relationship has transitioned to balanced development, while conflict relationships have generally decreased. All 16 Middle Eastern countries have joined the BRI, and China's cooperation model in the region has shifted to dual cooperation, with an increasing number of cooperative countries. Negative news between China and Middle Eastern countries has decreased, leading to a reduction in conflict relationships and diminished differences between countries.

China's interactions with the United Arab Emirates, Egypt, and Iran are dominated by cooperation, while interactions with Saudi Arabia, Oman, Jordan, and the remaining eight countries are balanced. It is important to maintain existing close relationships and cooperation momentum, promote high-quality development of the BRI, and consider project investments and cooperation based on the situation in conflict-dominated countries. Overall, China maintains active and varied interactions with Middle Eastern countries, aiming for mutual benefit, win-win cooperation, and enhanced economic development in the region.

While the GDELT database is powerful, it may have limitations such as insufficient event coverage, errors in news information coding, and unclear event categorization. Additionally, due to the continuous advancement of the Internet and the rapid development of various news media, the volume of published articles changes rapidly over time, leading to a "natural expansion" issue in the GDELT database's data size. In this study, data cleansing was performed before analysis. The impact degree of cooperation or conflict events between China and Middle Eastern countries was calculated using the Goldstein Scale and the number of event occurrences. The difference in impact degrees between cooperation and conflict events was used to represent the degree of bilateral relationships between countries, aiming to mitigate this issue. However, it cannot be completely eliminated, and the calculation process still requires further improvement.

Moreover, each event in the GDELT database contains multiple fields such as event type, tone of description, geographical information, etc. In the future, combining multiple methods to investigate the interaction between countries or regions can provide a more in-depth understanding, accurately identify key interactions, and better serve policy-making.

## Author Contributions

**Conceptualization:** Junhua Chen.

**Data curation:** Xiaolu Yang, Meijun Wang.

**Formal analysis:** Xiaolu Yang.

**Methodology:** Junhua Chen, Xiaolu Yang.

**Writing – original draft:** Junhua Chen, Xiaolu Yang.

**Writing – review & editing:** Junhua Chen, Xiaolu Yang, Meijun Wang, Min Su.

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
