## [Decision Letter · Decision Letter 0]

6 Oct 2023

PONE-D-23-26208Evolution of China's Interaction with Middle Eastern Countries under the Belt and Road InitiativePLOS ONE

Dear Dr. Yang,

Thank you for submitting your manuscript to PLOS ONE. After careful consideration, we feel that it has merit but does not fully meet PLOS ONE’s publication criteria as it currently stands. Therefore, we invite you to submit a revised version of the manuscript that addresses the points raised during the review process.

We look forward to receiving your revised manuscript.

Kind regards,

Masoud Behzadifar

Academic Editor

PLOS ONE

Journal Requirements:

4. We note that Figures 1, 3 and 4 in your submission contain map images which may be copyrighted. All PLOS content is published under the Creative Commons Attribution License (CC BY 4.0), which means that the manuscript, images, and Supporting Information files will be freely available online, and any third party is permitted to access, download, copy, distribute, and use these materials in any way, even commercially, with proper attribution. For these reasons, we cannot publish previously copyrighted maps or satellite images created using proprietary data, such as Google software (Google Maps, Street View, and Earth). For more information, see our copyright guidelines: http://journals.plos.org/plosone/s/licenses-and-copyright.

1.) You may seek permission from the original copyright holder of Figures 1, 3 and 4 to publish the content specifically under the CC BY 4.0 license.  

2.) If you are unable to obtain permission from the original copyright holder to publish these figures under the CC BY 4.0 license or if the copyright holder’s requirements are incompatible with the CC BY 4.0 license, please either i) remove the figure or ii) supply a replacement figure that complies with the CC BY 4.0 license. Please check copyright information on all replacement figures and update the figure caption with source information. If applicable, please specify in the figure caption text when a figure is similar but not identical to the original image and is therefore for illustrative purposes only.

Reviewers' comments:

Reviewer's Responses to Questions

**Comments to the Author**

1. Is the manuscript technically sound, and do the data support the conclusions?

Reviewer #1: Partly

Reviewer #2: Yes

2. Has the statistical analysis been performed appropriately and rigorously? 

Reviewer #1: Yes

Reviewer #2: Yes

3. Have the authors made all data underlying the findings in their manuscript fully available?

Reviewer #1: Yes

Reviewer #2: Yes

4. Is the manuscript presented in an intelligible fashion and written in standard English?

Reviewer #1: Yes

Reviewer #2: No

5. Review Comments to the Author

Reviewer #1: Overall, this article provides an intriguing analysis of the study using the Goldstine index. Here are a few points to consider for further improvement:

1- In the introduction section, in addition to the importance of the Middle East, it is better to refer to relations

China's international and strategic relationship with other regions of the world in the past and present is also mentioned, this will engage the reader's mind more to understand your study.

2- Although the Goldstine index is briefly mentioned in the methodology section, It is better to refer to the use of this index in previous studies. providing clear examples of the indexs application would be helpful. (more emphasis on explaining its parameters will enhance the validity and reliability of the analysis )

3- To enhance the clarity and conciseness of the discussion section, I suggest the author critically each reference. in this case, adding references based on similar studies can be valuable.

Reviewer #2: Thanks to the corresponding author and research team. It is clear that a lot of effort has been put into writing the manuscript.

1. Based on Formula “Interactive Relationship Types under Reciprocity”, the relationship between Iran and China; what is the position of ? be explained?

=1

 > 0.01

 < -0.01

2. Page 11; line 230 refers to Iran-China relations in political, diplomacy, economy, culture, and other fields, which has led to multi-level cooperation? The question that arises is that; Considering the many cultural and religious differences between the two countries, how were they able to overcome the differences and form multi-level cooperation?

3. Considering that China adheres to non-interference policy in the internal affairs of other countries; with what policy did he help to improve relations between Iran and Saudi Arabia? So that he can finally transform relations from the "single cooperation" approach to the "double cooperation" model?

6. PLOS authors have the option to publish the peer review history of their article (what does this mean?). If published, this will include your full peer review and any attached files.

Reviewer #1: No

Reviewer #2: **Yes: **zeynab farhadi

Social Determinants of Health Research Center, Health Research Institute, Babol University of Medical Sciences, Babol, Iran

---

## [Author Response · Author response to Decision Letter 0]

9 Oct 2023

Dear Dr. Behzadifar and Reviewers,

Thank you for your insightful comments and suggestions. We appreciate the time and effort you have invested in reviewing our manuscript.

Reviewer 1:

1. In the introduction section, in addition to the importance of the Middle East, it is better to refer to relations China's international and strategic relationship with other regions of the world in the past and present is also mentioned, this will engage the reader's mind more to understand your study.

We sincerely thank the reviewer for careful reading. As suggested by the reviewer, we have incorporated a discussion on China's international strategies and developmental processes across various regions worldwide in the opening of this article.

2. Although the Goldstine index is briefly mentioned in the methodology section, It is better to refer to the use of this index in previous studies. providing clear examples of the indexs application would be helpful. (more emphasis on explaining its parameters will enhance the validity and reliability of the analysis ) 

This is a valuable suggestion, and we have supplemented information on the use cases of Goldstein scores in the article, specifically at line 143. Additionally, we have included references 36 and 37 to support this update.

[36] Li, YH; Jian, Z; Zhao, LX. How political conflicts distort bilateral trade: Firm-level evidence from China. Journal of Economic Behavior & Organization. 2021;183, 233-249

[37] Zhang, X., Zhang, X., Zhang, L., et al. Spatial-temporal Evolution Characteristics of China's Geopolitical Relations with Belt and Road Countries. World Geography Research, 2023;32(09), 17-27.

3. To enhance the clarity and conciseness of the discussion section, I suggest the author critically each reference. in this case, adding references based on similar studies can be valuable.

Thank you for your valuable suggestions. Given the multitude of countries involved in this study and the limited existing research, I have made concerted efforts to comprehensively reorganize the discussion section. Additionally, I have included references 41, 42, and 43 to further support and enrich the context. Your insights have been instrumental in refining the content of the paper.

[41] YU Zhen. Third-Party Market Cooperation between China and Middle Eastern Countries under the Background of the Belt and Road Initiative. International Relations Studies, 2023;2023(04), 59-81+157-158.

[42] Chaziza, M. China's Economic Diplomacy Approach in the Middle East Conflicts. China Report. 2019;55 (1), pp.24-39

[43] Almujeem, NS. GCC countries' geoeconomic significance to China's geopolitical ends. Review Of Economics And Political Science. 2021;6(4), 348-363

Reviewer 2:

1. Based on Formula “Interactive Relationship Types under Reciprocity”, the relationship between Iran and China; what is the position of 𝑅𝑗? be explained?

We apologize for not clearly stating this concept in the article. Let 𝑅𝑗 represent the type of interaction between China and Iran. When 𝑅𝑗 > 0.01, it indicates that the impact of cooperative events between China and Iran, as a proportion of the total impact of China's cooperation events in the Middle East, significantly exceeds the proportion of conflict events' impact. In this scenario, the interaction between China and Iran is classified as cooperation-dominant. Conversely, when 𝑅𝑗 < -0.01, China's interaction with Iran is conflict-dominant, meaning the impact of conflict events surpasses that of cooperative events. For cases where -0.01 ≤ 𝑅𝑗 ≤ 0.01, the interaction between China and Iran is considered balanced.

Over the decade from 2013 to 2022, more than half of the Middle Eastern countries demonstrated a cooperative-dominant interaction with China. As depicted in Figure 5, China and Iran exhibited a cooperation-dominant relationship for six out of the ten years, with 𝑅𝑗 greater than 0.01, and conflict-dominant for four years with 𝑅𝑗 less than -0.01. Consequently, over the past decade, the interaction between Iran and China, in comparison to the overall interactions in the Middle East, tends to be characterized as cooperation-dominant in this study.

2. Page 11; line 230 refers to Iran-China relations in political, diplomacy, economy, culture, and other fields, which has led to multi-level cooperation? The question that arises is that; Considering the many cultural and religious differences between the two countries, how were they able to overcome the differences and form multi-level cooperation?

Thank you for your thoughtful question. We sincerely apologize if any part of the statement lacked clarity. The enduring history of amicable interactions between Iran and China, despite cultural and religious distinctions, has been characterized by continuous economic collaboration. In recent years, propelled by international dynamics, the political and diplomatic ties between our two nations have notably strengthened, forming a robust alliance across various fronts. To eliminate any potential misunderstandings, I have carefully revised ambiguous expressions in the original text. Your inquiry is greatly appreciated.

3. Considering that China adheres to non-interference policy in the internal affairs of other countries; with what policy did he help to improve relations between Iran and Saudi Arabia? So that he can finally transform relations from the "single cooperation" approach to the "double cooperation" model?

We appreciate your attention to the matter. Regarding China's influence on the improved relations between Iran and Saudi Arabia, it's crucial to note that the primary driver for reconciliation between these two nations arises from their intrinsic willingness to mend ties.

Examining the respective situations in Iran and Saudi Arabia, the Middle East dynamics, and the global landscape, the years spanning from 2016 to 2020 were not conducive to a resolution due to divergent positions in conflicts such as Syria and Yemen, as well as varied approaches to addressing the "Islamic State" threat. However, starting in 2020, both parties embarked on a gradual path towards easing tensions, culminating in a nearing reconciliation in 2023.

China, upholding a policy of non-interference in the internal affairs of sovereign states, emerged as an external mediator acceptable to both sides. This diplomatic role has effectively facilitated the improvement of bilateral relations. Furthermore, as a major global consumer market and a proponent of the Belt and Road Initiative, China has engaged in cooperative efforts with Iran and Saudi Arabia.

Please find the revised manuscript attached. We believe these changes strengthen the overall quality of the paper.

Thank you once again for your valuable feedback.

Sincerely,

Yang Xiaolu

yxl5011@email.swu.edu.cn

---

## [Editor Report · Decision Letter 1]

12 Oct 2023

Evolution of China's Interaction with Middle Eastern Countries under the Belt and Road Initiative

PONE-D-23-26208R1

Dear Dr. Yang,

We’re pleased to inform you that your manuscript has been judged scientifically suitable for publication and will be formally accepted for publication once it meets all outstanding technical requirements.

Kind regards,

Masoud Behzadifar

Academic Editor

PLOS ONE
---

## [Editor Report · Acceptance letter]

31 Oct 2023

PONE-D-23-26208R1 

Evolution of China's Interaction with Middle Eastern Countries under the Belt and Road Initiative 

Dear Dr. Yang:

I'm pleased to inform you that your manuscript has been deemed suitable for publication in PLOS ONE. Congratulations! Your manuscript is now with our production department. 

Kind regards, 

on behalf of

Dr. Masoud Behzadifar 

Academic Editor

PLOS ONE